# Boron-mediated directed aromatic C–H hydroxylation

Jiahang Lv[1,2], Binlin Zhao[1], Yu Yuan[2], Ying Han[2] & Zhuangzhi Shi [1✉]

Transition metal-catalysed C–H hydroxylation is one of the most notable advances in synthetic chemistry during the past few decades and it has been widely employed in the preparation of alcohols and phenols. The site-selective hydroxylation of aromatic C–H bonds under mild conditions, especially in the context of substituted (hetero)arenes with diverse functional groups, remains a challenge. Here, we report a general and mild chelation-assisted C–H hydroxylation of (hetero)arenes mediated by boron species without the use of any transition metals. Diverse (hetero)arenes bearing amide directing groups can be utilized for ortho C–H hydroxylation under mild reaction conditions and with broad functional group compatibility. Additionally, this transition metal-free strategy can be extended to synthesize C7 and C4-hydroxylated indoles. By utilizing the present method, the formal synthesis of several phenol intermediates to bioactive molecules is demonstrated.

[1] State Key Laboratory of Coordination Chemistry, Chemistry and Biomedicine Innovation Center (ChemBIC), School of Chemistry and Chemical Engineering, Nanjing University, Nanjing, China. [2] College of Chemistry and Chemical Engineering, Yangzhou University, Yangzhou, China. ✉email: shiz@nju.edu.cn

Phenols are structural constituents of pharmaceuticals, agrochemicals, polymers, and naturally occurring compounds and serve as versatile synthetic intermediates[1–4]. Bioactive molecules of particular interest are (hetero)arenes such as amides, indolines, and indoles-containing hydroxyl groups (Fig. 1a)[5–11]. The site-selective introduction of a hydroxyl group to a (hetero)arene is an important task in both chemical industry and organic synthesis. Traditional methods used for phenol preparation include nucleophilic aromatic substitution of activated aryl halides[12] and Sandmeyer-type hydroxylation[13], as well as the transition-metal-catalysed hydroxylation of (hetero)aryl halides with hydroxide salts (e.g., KOH and NaOH)[14–18], all of which require the presence of a (pseudo)halide in the (hetero) arenes. During the past decade, C–H functionalization has gained considerable momentum, holding great promise for avoiding the preinstalled functional groups[19–28]. Among these C–H functionalization techniques, hydroxylation is one of the most important C–H functionalization reactions[29,30]. As early as 1990, Fujiwara et al. explored the hydroxylation of benzene using $O_2$ as the oxidant enabled by Pd catalysis[31]. However, this pioneering work had several limitations, such as a low efficiency, poor selectivity, and harsh reaction conditions. Substrates bearing a chelating functional group can coordinate with the metal catalyst and undergo further C–H functionalization[32–34]. In this context, several groups have explored transition-metal-catalysed directed aromatic C–H hydroxylation using organic oxidants, hydrogen peroxide or molecular oxygen (Fig. 1b)[35–46]. While synthetically very attractive, most of these protocols still suffer from the use of expensive noble metals, such as Pd, Rh, Ru, and Ir, as catalysts. This requirement may be a significant limitation, especially for applications needing large-scale synthesis methods and for the removal of toxic trace metals from pharmaceutical products. From a synthetic perspective, the ability to prepare synthetically relevant scaffolds via regio-controlled C–H hydroxylation under mild conditions by using cheap oxidants and avoiding the use of transition metals would be of great importance.

The transition-metal-catalysed C–H borylation reaction has emerged as an effective method for the construction of arylboronic acids and their derivatives[47–49]. Recently, our group[50] and the Ingleson group[51] reported a general strategy for the mild directed C–H borylation of (hetero)arenes using $BBr_3$ as both the reagent and catalyst under metal-free conditions[52–55]. $BBr_3$ is an attractive borylation agent because it is a commercially available in multigram to kilogram quantities and is cheaper than most common boron reagents. To further extend the utility of this strategy, here, we developed a one-pot directed C–H borylation/ oxidation protocol to access numerous structurally diverse phenols, whose regioselectivity is not easily accessed by traditional strategies (Fig. 1c). Replacing the transition-metal-catalysed C–H hydroxylation process by a boron-mediated strategy offers an

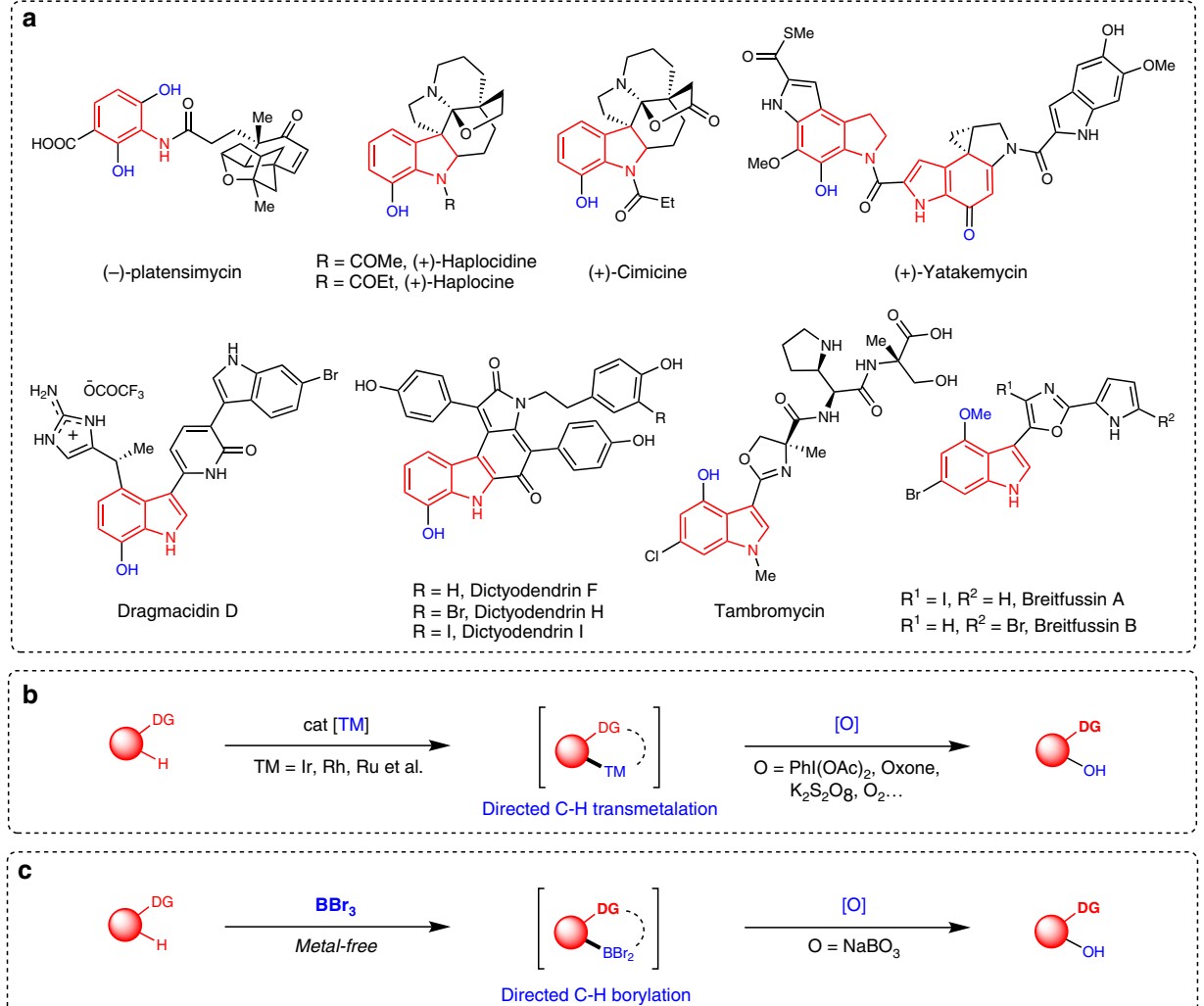

**Fig. 1 Towards a transition-metal-free process for directed aromatic C–H hydroxylation. a** Phenol-based bioactive molecules. **b** Transition-metal-catalysed directed aromatic C–H hydroxylation. **c** Our approach for directed aromatic C–H hydroxylation under transition-metal-free conditions.

**Table 1 Optimization of the reaction conditions.[a]**

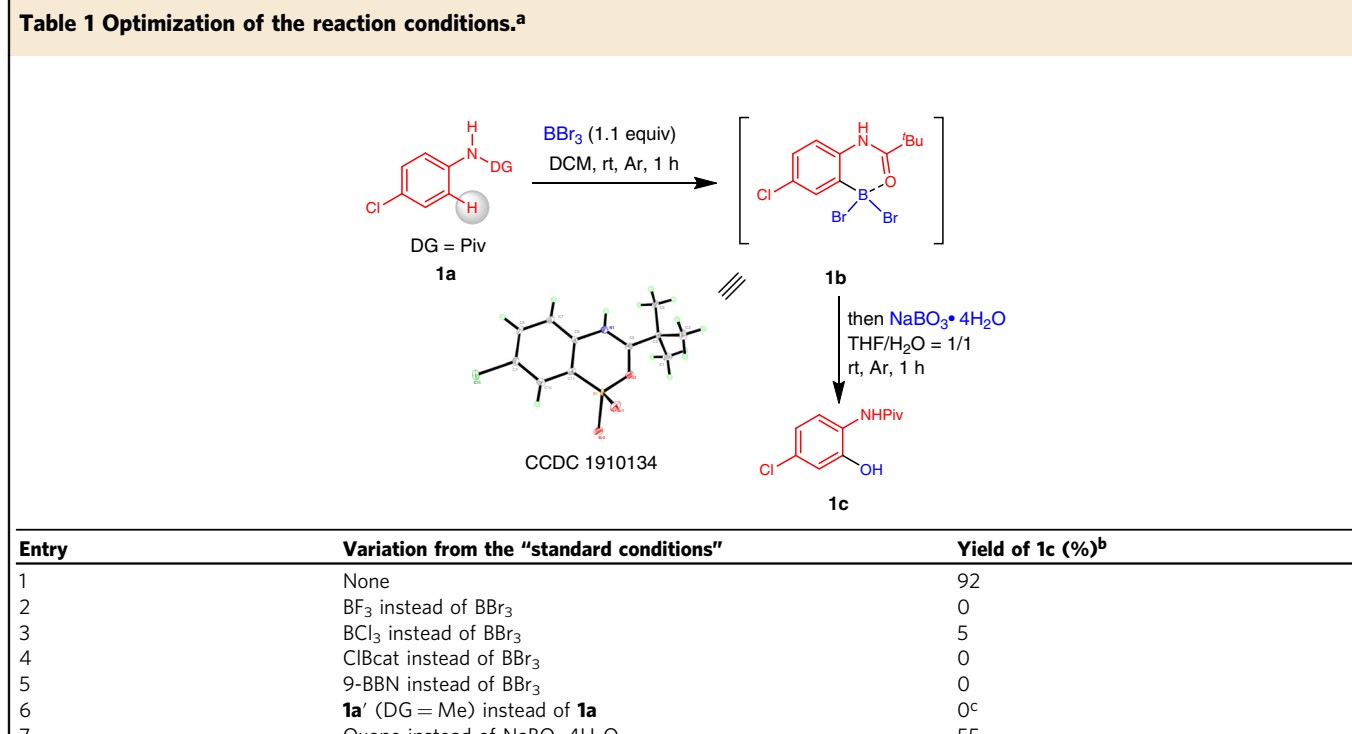

CCDC 1910134

| Entry | Variation from the "standard conditions" | Yield of 1c (%)[b] |
|---|---|---|
| 1 | None | 92 |
| 2 | BF₃ instead of BBr₃ | 0 |
| 3 | BCl₃ instead of BBr₃ | 5 |
| 4 | ClBcat instead of BBr₃ | 0 |
| 5 | 9-BBN instead of BBr₃ | 0 |
| 6 | 1a' (DG = Me) instead of 1a | 0[c] |
| 7 | Oxone instead of NaBO₃·4H₂O | 55 |
| 8 | H₂O₂ instead of NaBO₃·4H₂O | 73 |

[a]Standard conditions: **1a** (0.20 mmol), BBr₃ (0.22 mmol) in 1.0 mL of DCM at room temperature, 1 h, under Ar; then, 1 mL of THF/H₂O (1/1) and NaBO₃·4H₂O (0.6 mmol) were added to the mixture at room temperature, 1 h, under Ar.
[b]Isolated yield.
[c]The corresponding *ortho* C–H borylation product.

alternative pathway for synthesizing phenols and has exciting possibilities because of the superior practicality, low cost, and environmental friendliness of this alternate method.

## Results

**Reaction design**. We initiated our study by investigating the reaction of *N*-pivaloyl amide **1a** with BBr₃ (Table 1). As a result, we discovered that the use of 1.0 equivalent of **1a** with 1.1 equivalents of BBr₃ in DCM at room temperature for 1 h led to the full conversion of the precursors and formation of boron complex **1b**. Then, the treatment of 3.0 equivalents of NaBO₃ in a THF and H₂O (1:1) co-solvent led to the in situ formation of the hydroxylated product, and phenol **1c** was isolated with a 92% yield (Table 1, entry 1). Other boron halides, such as BF₃, were not efficient for this reaction (Table 1, entry 2), and BCl₃ only afforded a trace amount of the product (Table 1, entry 3). When the reaction was carried out using ClBcat or 9-BBN, we did not observe any C–H borylation or hydroxylation products (Table 1, entries 4–5). The substrate **1a'** bearing an N-Me group failed to achieve this transformation, confirming the importance of the *N*-pivaloyl moiety for achieving both a high reactivity and selectivity (Table 1, entry 6). To our delight, other common oxidants, such as oxone and H₂O₂, were also effective for this hydroxylation process, generating the desired product **1c** in slightly lower yields (Table 1, entries 7–8).

**Scope of the methodology**. We first examined the scope of the *ortho*-selective C–H hydroxylation of amides (Fig. 2). When the simple *N*-pivaloyl amide **2a** was employed as a substrate, hydroxylation proceeded at the *ortho* C–H bond, affording **2c** with a 85% yield. Amides bearing methyl (**3–5a**), *t*Bu (**6a**), phenyl (**7a**), and halogen-containing motifs (**8–13a**) at the *ortho*, *meta*,

and *para* positions underwent facile hydroxylation and afforded the corresponding products **3–13c** in good to excellent yields. The amides bearing electron-withdrawing groups such as CF₃ (**14–15a**), COOMe (**16a**), and CN (**17a**) are particular noteworthy; these substrates produced *ortho*-hydroxylated products **14–17c** with 66–80% yields. Electron-donating groups such as OTBS (**18c**) and SMe (**19c**) at the *para* position of the amides are tolerated. Substrate **20a** bearing a methoxy group can undergo *ortho* C–H hydroxylation and *O*-demethylation to generate the corresponding product **20c** with a 81% yield. Notably, the phenyldiazenyl substituent in substrate **21a**, which is also susceptible to C–H borylation, remained intact during the reaction. Other *N*-pivaloyl amides, including *N*-methylaniline (**22c**), tetrahydroquinoline (**23c**), and indoline (**24c**), are also tolerated for C–H hydroxylation. This protocol is compatible with heterocyclic motifs such as thiophene **25c**. Polyaromatic substrates **26–28c** were also shown to be highly reactive. As a prominent structural motif, *N*-arylpyrrolidinones have been used in Ru(II)-catalysed C–H hydroxylation[56,57]. We found that the boron-mediated directed C–H hydroxylation of *N*-phenylpyrrolidinone (**29a**) in the presence of BBr₃ could provide the desired product **29c** with a 79% yield. The system was compatible with the different *para*- and *meta*-substitution patterns in the phenyl ring of the N-arylpyrrolidinone backbone (**30–36c**). In addition, this C–H hydroxylation method is not limited to *N*-arylpyrrolidinones. Lactams such as **37–38a**, oxazolidin-2-one **39a**, and thiophene **40a** could also undergo C–H hydroxylation at the *ortho* position, affording good yields of products **37–40c**. Subjecting *N*-pivaloyl amides **41–45a**, which are substrates bearing two *N*-pivaloyl directing bonds, to our system resulted in the selective formation of the difunctionalization products **41–45c** in 60–89% yields. These bisphenols could be utilized as precursors for construction of polymers[58].

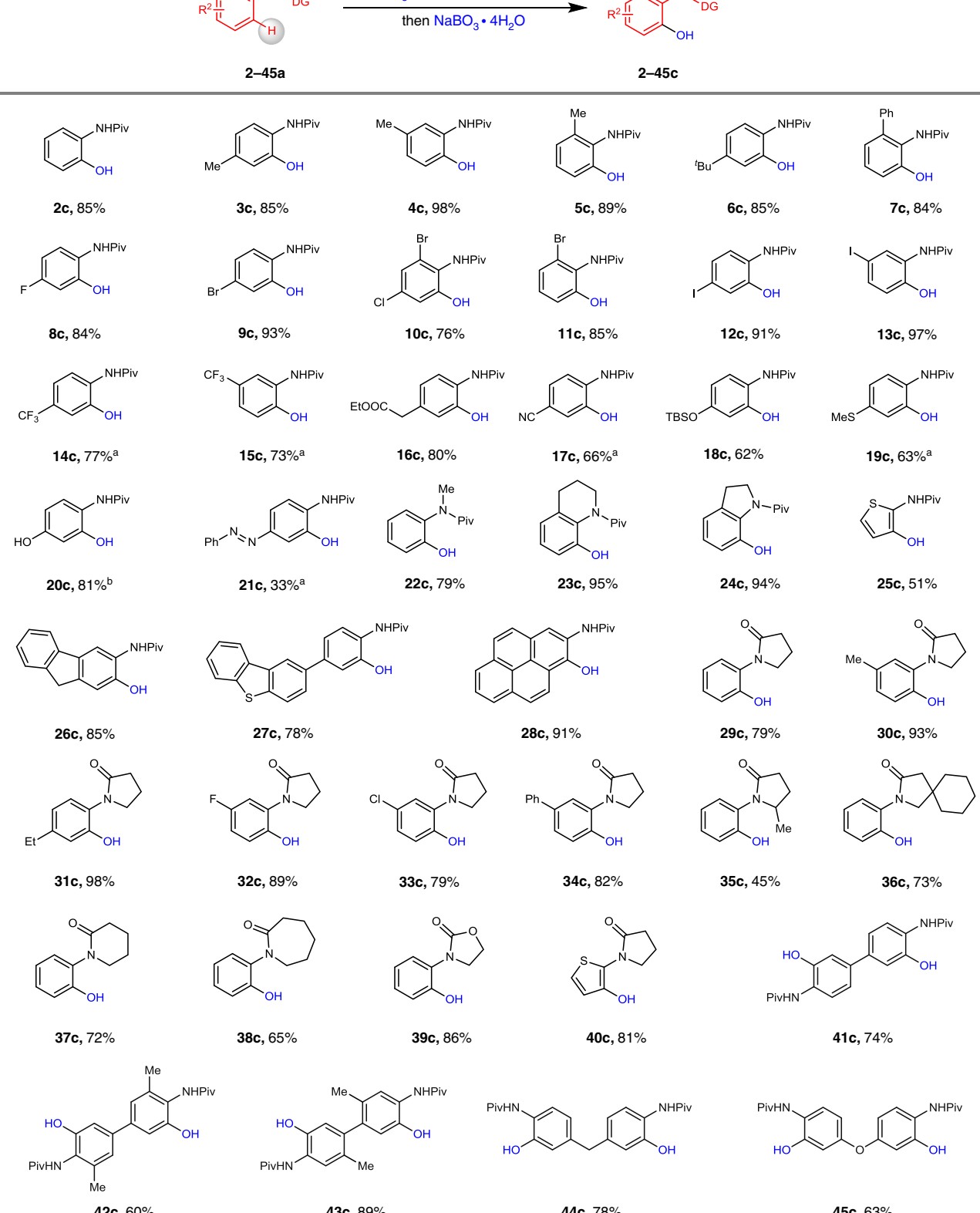

**Fig. 2 Boron-mediated directed *ortho* C–H hydroxylation of amides.** Reaction conditions: substrates **1a–28a** (0.20 mmol), BBr₃ (0.22 mmol) in 0.5 mL DCM at room temperature, 1 h, under Ar; NaBO₃·4H₂O (0.60 mmol) in 0.5 mL THF and 0.5 mL H₂O, at room temperature, 1 h. **29a–40a** (0.20 mmol), BBr₃ (0.60 mmol) in 0.5 mL DCM at 60 °C, 24 h; NaBO₃·4H₂O (0.60 mmol) in 0.5 mL THF and 0.5 mL K₂CO₃ (aq), at room temperature, 1 h. **41a–45a** (0.20 mmol), BBr₃ (0.40 mmol) in 0.5 mL DCM at room temperature, 1 h, under Ar; NaBO₃·4H₂O (1.50 mmol) in 0.5 mL THF and 0.5 mL H₂O, at room temperature, 1 h. ᵃUsing BBr₃ (2.0 mmol) in 0.1 mL of DCM. ᵇN-(4-methoxyphenyl)pivalamide (0.20 mmol), BBr₃ (0.5 mmol).

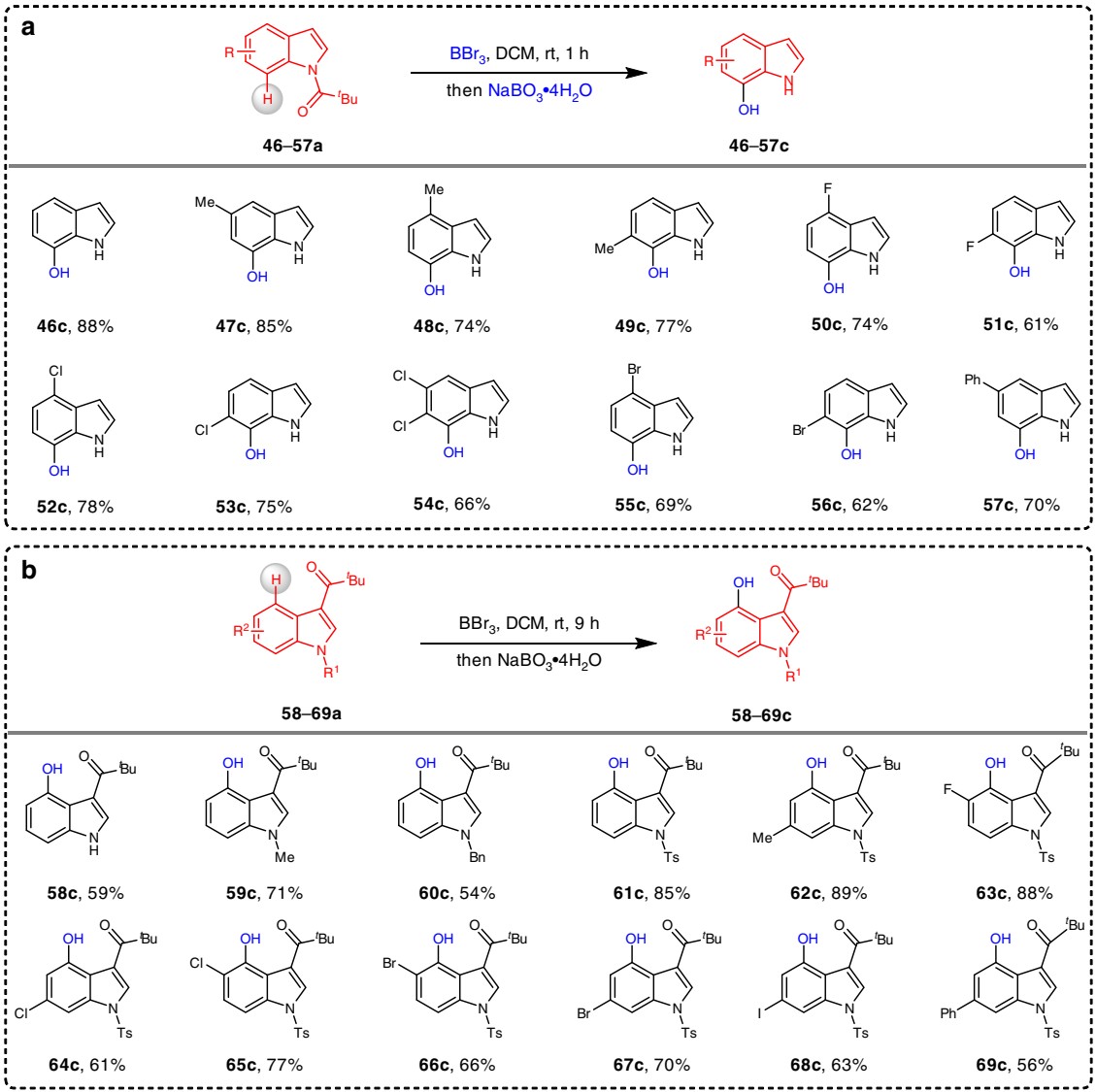

**Fig. 3 Boron-mediated directed C–H hydroxylation of indoles. a** Directed C–H hydroxylation of indoles at the C7 position. **b** Directed C–H hydroxylation of indoles at the C4 position. Reaction conditions: substrates **46–57a** (0.20 mmol), BBr$_3$ (0.22 mmol) in 0.5 mL DCM at room temperature, 1 h, under Ar; NaBO$_3$·4H$_2$O (0.60 mmol) in 0.5 mL THF and 0.5 mL K$_2$CO$_3$ (aq), at room temperature, 1 h; **58–60a** (0.20 mmol), BBr$_3$ (0.60 mmol) in 0.5 mL DCM at 60 °C, 10 h, under Ar; NaBO$_3$·4H$_2$O (1.50 mmol) in 0.5 mL THF and 0.5 mL H$_2$O, at 60 °C, 6 h; **61–69a** (0.20 mmol), BBr$_3$ (0.22 mmol) in 0.5 mL DCM at room temperature, 9 h, under Ar; NaBO$_3$·4H$_2$O (1.0 mmol) in 0.5 mL THF and 0.5 mL H$_2$O, at room temperature, 2 h.

We next investigated the scope of the C7 selective C–H hydroxylation of indoles (Fig. 3a). We found that indole **46a** could generate 7-hydroxyindole **46c** with a 88% yield by a cascade C–H borylation/oxidation/DG removal protocol, in which the N-Piv group can be removed automatically during work-up with K$_2$CO$_3$. Indoles bearing methyl (**47–49a**) substituents at the 4–6 positions underwent facile hydroxylation and afforded the corresponding products in 74-85% yields. Again, halogen-containing motifs (F, Cl, and Br, **50–56a**) work very well in the C7 selective borylation process. In addition, substrate **57a** contains a phenyl substituent also delivering coupled product **57c** with a 70% yield. We further examined the scope of using C3-pivaloyl indoles as coupling partners with BBr$_3$; these compounds reacted with a high regioselectivity to produce C4-hydroxylated indoles (Fig. 3b)[59]. We first evaluated the influence of the N–H protection groups on the indoles. Notably, the free indole **58a** could provide the desired product **58c** with a 59% yield. The treatment of the indoles **59–60a** bearing N-Me and N-Bn groups in the system provided a 71% and 54% isolated yields of the

corresponding C4-hydroxylation products **59–60c**. Indole **61a** bearing an N-Ts protection group can promote the reactivity of this transformation, affording the product **61c** with a 85% yield. Regarding the scope of the indole framework, diverse substituents, including methyl (**62c**), F (**63a**), Cl (**64–65c**), Br (**66–67c**), I (**68c**), and phenyl (**69c**) are tolerated.

**Synthetic applications**. To further demonstrate the potential synthetic applications of this method, we showed three examples to compare existing strategies with our developed C–H hydroxylation method. Previously, using N-acetylindoline **70** as a model substrate for the total synthesis of the potent caspase-8 inhibitor (+)-haplocidine and its N1-amide congener (+)-haplocine, the precursor acetoxy-indoline **71'** was generated with a 84% yield by the palladium-catalysed C7 hydroxylation of indoline[60]. Based on the boron-mediated strategy, we prepared product **71** from substrate **70** with a 71% yield, in which *N*-acetyl can be used as a directing group (Fig. 4a). Trauner and co-workers[61] reported the evolution of the total synthesis of exiguamines,

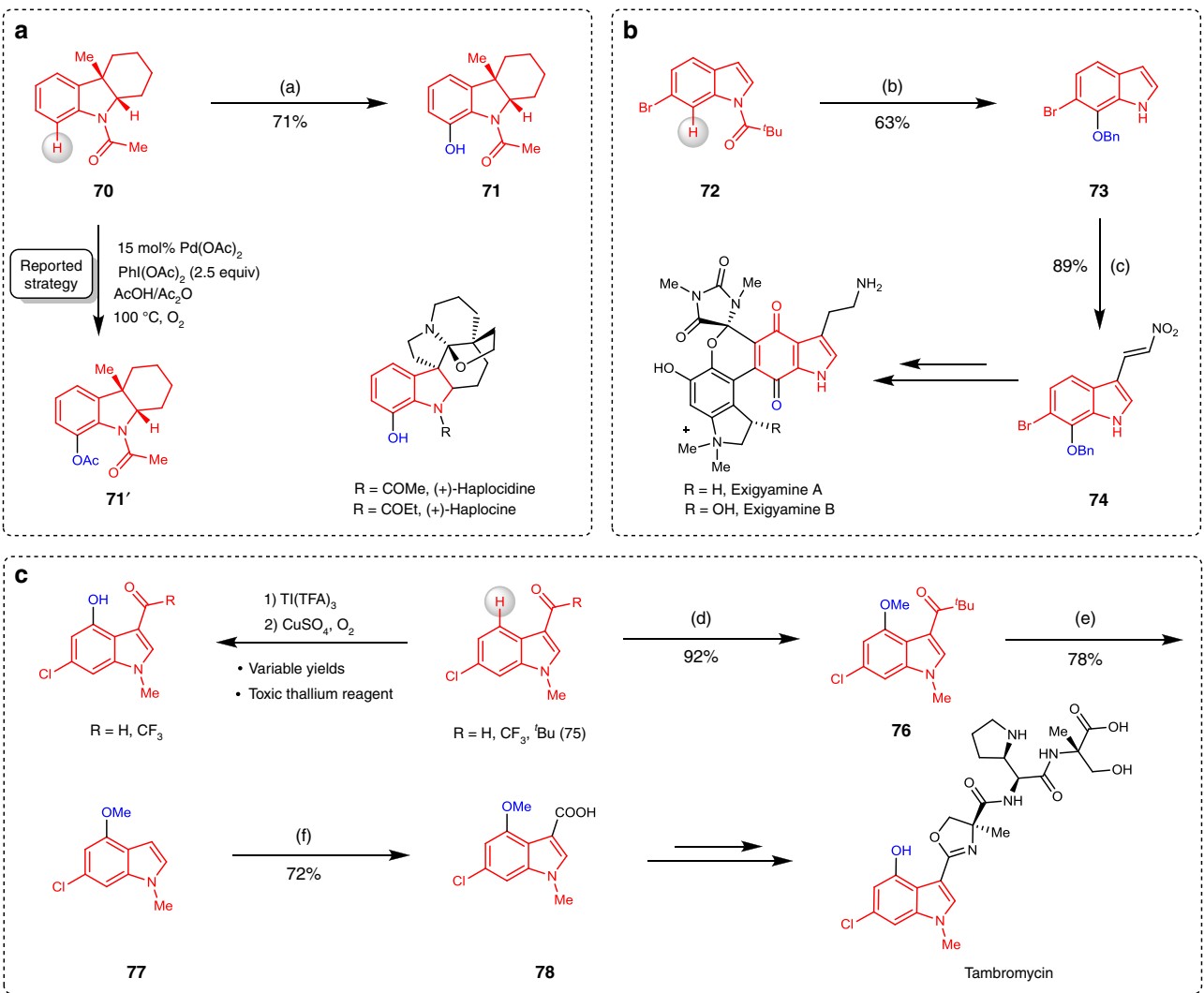

**Fig. 4 Synthetic applications. a** Using *N*-acylindoline **70** as a model substrate for the synthesis of (+)-haplocidine and (+)-haplocine. **b** Synthesis of the key intermediate **73** for the synthesis of the exiguamines. **c** Synthesis of the key intermediate **78** for the synthesis of tambromycin. Reagents and conditions: (a) **70** (0.2 mmol) and BBr$_3$ (0.6 mmol) in 0.5 mL DCM at 110 °C, 24 h; NaBO$_3$·4H$_2$O (1.0 mmol) in 0.5 mL of THF and 0.5 mL of sat. K$_2$CO$_3$, at 60 °C, 6 h. (b) **72** (0.2 mmol) and BBr$_3$ (0.22 mmol) in 0.5 mL of DCM at 25 °C, 1 h; NaBO$_3$·4H$_2$O (0.6 mmol) in 0.5 mL of THF and 0.5 mL of K$_2$CO$_3$ (aq) at 25 °C, 1 h; K$_2$CO$_3$ (0.6 mmol) and BnBr (0.24 mmol) in 2.0 mL of acetone at 25 °C, 24 h; (c) **73** (0.2 mmol) and POCl$_3$ (0.25 mmol) in 2.0 mL of dry DMF, reflux at 160 °C; NH$_4$OAc (0.22 mmol) in 1.0 mL MeNO$_2$, reflux at 115 °C; (d) **75** (0.2 mmol) and BBr$_3$ (0.6 mmol) in 0.5 mL of DCM, at 60 °C, 6 h; NaBO$_3$·4H$_2$O (2.0 mmol) in 0.5 mL of THF and 0.5 mL of K$_2$CO$_3$ (aq) at 25 °C, 1 h; NaH (0.24 mmol) in 1.0 mL of THF and MeI (0.24 mmol) at 25 °C, 1 h; (e) **76** (0.2 mmol), TsOH (0.3 mmol), and ethylene glycol (1,6 mmol) in 2.0 mL of toluene at 120 °C, 22 h; (f) **77** (0.2 mmol) and POCl$_3$ (0.25 mmol) in 2.0 mL of dry DMF, reflux at 160 °C; 2-methylbut-2-ene (2.6 mmol) in 3 mL of $^t$BuOH, NaClO$_2$ (0.74 mmol), NaH$_2$PO$_4$ (1.0 mmol) at 25 °C, 24 h.

where nitrovinylindole **74** was a key intermediate. To simplify this synthesis process, we provided an alternative route to **74** using the developed C–H hydroxylation protocol. The indole substrate **72** was regio-selectively hydroxylated at the C7 position and further deprotected and then protected as a benzyl ether to yield 7-hydroxy-6-bromoindole derivative **73** with a 63% yield. Then, indole **73** was formylated and condensed with nitro-methane to yield nitrovinylindole **74** with a 89% yield (Fig. 4b). The Renata group[62] recently identify a concise synthetic route to access tambromycin. During the study, they were drawn to a thallium-mediated C–H hydroxylation of indoles at the C4 position, suffering from highly variable yields and a lack of scalability. Inspired by this result, we finally focused our attention on the boron-mediated strategy to synthesize indole **78**. Using N-methyl indole **75** as a substrate, C4-hydroxylation was identified as a viable approach to access the desired indole fragment **76** after

etherification with MeI. To our delight, the removal of a pivaloyl group from **76** was readily accomplished by a reverse Friedel-Crafts reaction in the presence of TsOH and glycol, providing a good yield of **77**. Further C3 formylation and oxidation could provide a good yield of the key building block **78**, which was facile to convert to tambromycin (Fig. 4c).

## Discussion

In summary, we have developed an efficient boron-mediated system that is capable of mimicking the chelation-assisted metallic system to achieve directed C–H hydroxylation. The use of this method for the preparation of substituted phenols and downstream-functionalized products showcases the strategic opportunity to use this strategy for the synthesis of biologically active compounds. The reaction provides a simple new bond disconnection protocol for constructing these motifs with

different regioselectivities and broader functional group compatibilities than existing methods.

## Methods

**General procedure for the synthesis of phenol 1c.** A flame-dried 25 mL Schlenk tube was flushed with argon, and $N$-pivaloyl amide **1a** (0.2 mmol, 1.0 equiv) and dry DCM (0.5 mL, 0.4 M) were introduced. A solution of $BBr_3$ (1.0 M in DCM, 0.22 mL, 1.1 equiv) was added slowly under an argon atmosphere. The mixture was stirred at room temperature for 1 h. After stirring, the solvent was removed under vacuum directly. $NaBO_3 \cdot 4H_2O$ (92.3 mg, 0.6 mmol, 3.0 equiv), 0.5 mL of THF, and 0.5 mL of $H_2O$ were sequentially added to the reaction mixture and stirred at room temperature for another 1 h (monitored by TLC). After that, the excess water was removed by filtration with $MgSO_4$ and then washed with EtOAc (10.0 mL × 3). The filtrate was collected, and the crude mixture was directly subjected to column chromatography on a silica gel, using petrol ether/EtOAc (10/1) as the eluent to give the desired product **1c** as a white solid (41.5 mg, 92%).

## Data availability

The authors declare that the data supporting the findings of this study are available within the article and Supplementary Information file or from the corresponding author upon reasonable request. The X-ray crystallographic coordinates for the structures reported in this study have been deposited at the Cambridge Crystallographic Data Centre (CCDC) under deposition numbers CCDC 1910134. These data can be obtained free of charge from The Cambridge Crystallographic Data Centre via www.ccdc.cam.ac.uk/data_request/cif.

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

## Acknowledgements

This study was supported by National Natural Science Foundation of China (Grant 21972064 and 21672097), the Excellent Youth Foundation of Jiangsu Scientific Committee (Grant BK20180007), and the "Innovation & Entrepreneurship Talents Plan" of Jiangsu Province.

## Author contributions

Z.S. conceived the concept, directed the project, and wrote the paper. J.L. and B.Z. performed the experiments. Y.Y. and Y.H. discussed the results.

## Competing interests

The authors declare no competing interests.
