## [Peer Review File · Nature Communications]

Reviewers' comments:

Reviewer #1 (Remarks to the Author):

In this manuscript, Shi and coworkers described an efficient boron-mediated system that is capable of mimicking the chelation-assisted metallic system to access directed C-H hydroxylation. The reported reaction is derived from the authors' recent study on metal-free directed sp²-C-H borylation (Nature 2019, 575, 336.). Diverse arenes bearing amide directing groups can be utilized for ortho C-H hydroxylation under mild reaction conditions, resulting in broad functional group compatibility. In addition, this strategy can be extended to synthesize C7 and C4-hydroxylated indoles. They also applied the present method as the key steps to the formal synthesis of many phenol intermediates in bioactive molecules. This work represents an intriguing example in directed C-H hydroxylation since all those previous reactions have been based on the transition metal catalysts. In this sense, this work bears sufficient conceptual novelty and application significance for publication in Nature Communications.

Nonetheless, some minor revisions are necessary, as listed below.

- 1) Different boron species were screened in Table 1. Did you compare the BBr₃ with BI₃, which may be better due to the improved reactivity.
- 2) Only DG = Me was investigated in Table 2, is it possible to get the C-H hydroxylation products by using amides bearing DG = Ac, Boc et al.
- 3) It seems that H₂O₂ is more environmental friendly than NaBO₃. Why not use such oxidant in all cases.
- 4) In addition to the amides, did you use phenyl pivalates to get ortho C-H hydroxylation.
- 5) Regarding the transformation from compound 76 to 78, they removed the Piv group to indole 77, which underwent further C3 formylation and oxidation to the carboxylic acid 78. Did they try the Baeyer-Villiger oxidation to transfer 76 to the corresponding ester directly.

Reviewer #2 (Remarks to the Author):

The ability to direct the adjacent functionalization of anilines under Lewis Acidic conditions represents a promising development in directed chemical reactions. In the report the intermediate boronates are transformed to phenols providing a widely applicable synthesis of diverse aminophenols. This novel reactivity is of broad interest both to the synthetic community as well as any scientists endeavoring to synthesize complex molecules for specific purposes. The importance of accessing phenols, particularly under orthogonal conditions to directed metalation, can not be overstated. This report has been well drafted and provides the context where the research discovery fits within the existing, related methods for phenol synthesis. Publish immediately.

Reviewer #1 (Remarks to the Author):

In this manuscript, Shi and coworkers described an efficient boron-mediated system that is capable of mimicking the chelation-assisted metallic system to access directed C-H hydroxylation. The reported reaction is derived from the authors' recent study on metal-free directed sp²-C-H borylation (Nature 2019, 575, 336.). Diverse arenes bearing amide directing groups can be utilized for ortho C-H hydroxylation under mild reaction conditions, resulting in broad functional group compatibility. In addition, this strategy can be extended to synthesize C7 and C4-hydroxylated indoles. They also applied the present method as the key steps to the formal synthesis of many phenol intermediates in bioactive molecules. This work represents an intriguing example in directed C-H hydroxylation since all those previous reactions have been based on the transition metal catalysts. In this sense, this work bears sufficient conceptual novelty and application significance for publication in Nature Communications.

Response: I would thank you for the constructive comments, which were very helpful for us to improve the quality of this article.

Nonetheless, some minor revisions are necessary, as listed below.

1) Different boron species were screened in Table 1. Did you compare the BBr₃ with BI₃, which may be better due to the improved reactivity.

Response: Thanks for this kind suggestion. According to the reactivity, I agree that BI₃ may be better than BBr₃. We chose BBr₃ as the borylation agent because it is commercially available in multigram to kilogram quantities. Although BI₃ is also commercially available, the price of BI₃ is 100 times that of BBr₃.

2) Only DG = Me was investigated in Table 2, is it possible to get the C-H hydroxylation products by using amides bearing DG = Ac, Boc et al.

Response: Thanks for this kind suggestion. The substrate bearing Ac was compatible. You can see that compound **70** with DG = Ac can afford the desired product in 71% yield. However, the substrate bearing Boc failed at the current reaction conditions.

3) It seems that H₂O₂ is more environmental friendly than NaBO₃. Why not use such oxidant in all cases.

Response: Thanks for this kind suggestion. We provide such information of H₂O₂ in Table 1, entry 8. We didn't use H₂O₂ for all cases because 1) much lower yield was generated; 2) High concentration of H₂O₂ is explosive. Compared to H₂O₂, NaBO₃ is commercially available, very

cheap and easy-to-handle.

4) In addition to the amides, did you use phenyl pivalates to get ortho C-H hydroxylation.

Response: Yes. Similar to our former results in Nature (2019, 575, 336.), phenyl pivalates were failed to get *ortho* C-H borylation and further hydroxylation.

5) Regarding the transformation from compound **76** to **78**, they removed the Piv group to indole **77**, which underwent further C3 formylation and oxidation to the carboxylic acid **78**. Did they try the Baeyer-Villiger oxidation to transfer **76** to the corresponding ester directly.

Response: Thanks for this kind suggestion. In fact, a great deal of effort was put into the Baeyer-Villiger oxidation of **76** to form ester of **78**. However, the best result was only 8% yield in GC-MS, which was much lower than the indirect method in the text.

Reviewer #2 (Remarks to the Author):

The ability to direct the adjacent functionalization of anilines under Lewis Acidic conditions represents a promising development in directed chemical reactions. In the report the intermediate boronates are transformed to phenols providing a widely applicable synthesis of diverse aminophenols. This novel reactivity is of broad interest both to the synthetic community as well as any scientists endeavoring to synthesize complex molecules for specific purposes. The importance of accessing phenols, particularly under orthogonal conditions to directed metalation, can not be overstated. This report has been well drafted and provides the context where the research discovery fits within the existing, related methods for phenol synthesis. Publish immediately.

Response: Thanks for such positive comments.